# Dementia: A call for a paradigm shift in pre-registration nurse education

Isaac Tuffour  and Griffin Ganga

Faculty of Health, Education, and Wellbeing, University of Wolverhampton, Wolverhampton, UK

## Overview Review

Dementia: A call for a paradigm shift in
pre-registration nurse education. *Cambridge
Prisms: Global Mental Health*, **11**, e2, 1–6

dementia; pre-registration; nursing; education;
United Kingdom

**Corresponding author:**
Isaac Tuffour;
Email: i.tuffour@wlv.ac.uk

## Abstract

Dementia is a progressive brain disorder that affects memory, thinking and behaviour. It is a major global public health concern, with an estimated 55 million people worldwide living with the condition. In the UK, there is an estimated 944,000 people with dementia. This number is expected to double by 2050. Dementia is a major cause of disability and dependency, and it places a significant burden on families and carers. The current level of dementia education in pre-registration nursing programmes in the UK is inadequate. There are no pre-registration nursing educational programmes that offer dementia as a speciality. This is a major concern, as nurses are the primary providers of care to people with dementia. This article argues that dementia should be established as a branch of pre-registration nursing education that leads to a Registered Nurse (RN) – Dementia. This could help to address the shortage of specialist dementia nurses in the country. This article provides an important suggestion for countries with a shortage of specialist dementia nurses to consider establishing a stand-alone pre-registration branch of dementia nurse education. This would result in a more specialised workforce with the skills and knowledge to provide high-quality care to people with dementia.

## Impact statement

Dementia has become a global public health concern, with a projected increase in its prevalence in the coming years. Yet, the availability of specialist nurses capable of delivering the complex care needed for individuals afflicted by dementia remains limited. There is a shortage of specialist trained dementia nurses in the UK. Moreover, many qualified nurses lack the requisite skills and knowledge to provide person-centred care to dementia patients. This shortcoming can be linked to the absence of comprehensive dementia-focussed curricula within the pre-registration nursing programmes. It is indeed surprising that, despite the escalating demands prompted by an ageing population and the mounting urgency for specialist nursing care amid the dementia epidemic, numerous inquiries and committees tasked with improving nursing education failed to recommend pre-registration nursing programmes that could lead to Registered Nurse (RN) – Dementia. This article advocates for the establishment of a dedicated pre-registration nursing programmes leading to Registered Nurse (RN) – Dementia to address the critical shortage of specialist dementia nurses. The establishment of pre-registration nursing programmes in dementia care would guarantee that nurses are equipped with the indispensable competencies to provide person-centred and preventive care. Ultimately, it is hoped that this article will serve as a catalyst for countries confronting a shortage of specialist dementia nurses to consider the implementation of a pre-registration nursing programmes specific to dementia care to alleviate these problems.

## Introduction

The World Health Organisation (WHO) recognises dementia as a global public health concern and has developed a comprehensive global action plan for governments around the world to implement policies to reduce the growing number of dementia cases and improve the lives of people with dementia and their carers (World Health Organisation (WHO), 2023). Recognising the significance of dementia as the population ages has resulted in a flurry of policy efforts and documents across the United Kingdom to improve the care for dementia patients. These efforts have emphasised the importance of collaborative and integrated working between health and social care professionals and agencies, as well as the importance of a well-trained dementia workforce (National Institute for Clinical Excellence and Social Care Institute for Excellence (NICE/SCIE); National Collaborating Centre for Mental Health (UK), 2006; Department of Health, 2009; Health Education England (HEE), 2018). Emphasis has also been placed on quality standards to improve outcomes of health and social care for people with dementia from well-trained staff (National Institute for Health and Care Excellence (NICE), 2010, 2019). However, a

nationwide audit of dementia care in 2011 showed that workers in healthcare settings frequently lacked the knowledge and prerequisite skills in dementia care (Royal College of Psychiatrists (RCPsych), 2011). Concerns have been raised about the standards of medical education (Tullo and Gordon, 2013) and pre-registration nursing education (Pulsford et al., 2007) in dementia care. The U.K. government has attempted to address these concerns by setting targets to increase the number of staff receiving dementia training (Department of Health, 2013). This was followed by a national framework outlining England's aspirations for educational content and learning outcomes for dementia Health Education England (2018), Furthermore, the Prime Minister's Challenge on Dementia 2020 (Department of Health, 2016) mentions the ambition for the UK to become a world leader in establishing standards for dementia care and research. Likewise, the NHS Long Term Plan (NHS, 2019) pledges to improve dementia care. Taken together, these are timely, exciting and forward-thinking policy initiatives, however, they ignore the elephant in the room and fail to recognise that there is a shortage of specialist trained nurses to care for people with dementia, and such deficit has the potential to derail these policy initiatives. Particularly, while the Prime Minister's Challenge on Dementia 2020 lays out 50 pledges under four themes to reduce risk, improve health and care, create awareness and social action and promote research in dementia (Department of Health, 2016), it does not mention professional nursing training as one of the initiatives to improve dementia care. Nurses bear a significant care burden for people with dementia, and it has been suggested that the standards of dementia education in nursing curricula are either inadequate or not standardised (Pulsford et al., 2007; Collier et al., 2015; Cariñanos-Ayala et al., 2022).

Despite policy promises to enhance the health of people with dementia throughout their illness trajectory (Hoe et al., 2023), inadequate dementia-focussed curricula in pre-registration nursing programmes are jeopardising such efforts (Alushi et al., 2015; Surr et al., 2017; Harrison-Dening, 2019). Although there is evidence that few nurses with specialist training are doing their best to provide person-centred and preventive care to reduce hospital admissions for people with dementia (Kings Fund, 2016), evidence suggests that there is a correlation between the level of specialist training and the ability to provide physical and mental health support to individuals with dementia and their caregivers (Royal College of Psychiatrists (RCPsych), 2016). It has been found that nurses who have not had proper dementia training may lack confidence and experience compassion fatigue (Huang et al., 2013; De Witt and Ploeg, 2016). This article seeks to remedy these problems by presenting the argument for dementia to be established as a branch of pre-registration nursing education that leads to a Registered Nurse (RN) – Dementia. We believe that given the shortage of specialist dementia nurses, our proposal represents a long overdue paradigm shift and advancement in nursing education. Even though this is a contested proposition, it excites us as nursing academics to be able to contribute to the Prime Minister's Challenge on Dementia 2020 (Department of Health, 2016), and the NHS Long Term Plan (NHS, 2019) goals to improve dementia care. Moreover, this article is inspired by our experience as nursing academics. Over the years, we have mostly received two constructive criticisms from our students assigned to dementia clinical placements. First, students believe they were not adequately exposed to dementia learning experiences prior to being allocated to this clinical area. Second, students who appreciate this field of clinical practice believe that dementia nursing is distinct and should be recognised as a specialty, like mental health, learning disability or adult nursing. Although we are optimistic about the potential of developing specialist nurses to improve dementia care, we acknowledge that this may not be sufficient to address the challenges in dementia care. The development of specialist nurses must be part of a wider effort to provide better and integrated care that prioritises the development of dementia-friendly hospitals and care facilities, as well as a specialist interdisciplinary and multidisciplinary workforce (Department of Health, 2016). We are encouraged by the recent publication of the 'NHS Long Term Workforce Plan', which aims to take significant steps to reform how future nurses and NHS workforce are trained (NHS England, 2023). However, the plan has been criticised for lacking detail on how to retain nurses and achieve ambitious targets to recruit student nurses (Church, 2023).

## Dementia: An overview

Dementia is a broad term that encompasses group of diseases that are typically chronic or progressive, affecting multiple cognitive functions including memory, thinking, and behaviour, and significantly impair a person's ability to carry out everyday activities (American Psychiatric Association, 2013; Prince et al., 2014; Dixon et al., 2022; Senanayake, 2022). Cognitive impairment is frequently accompanied, and occasionally preceded, by changes in mood, emotional control, behaviour or motivation. Alzheimer's disease is the most common type of dementia, accounting for 60–70% of cases (Machado et al., 2021; Sargunam, 2021; World Health Organisation (WHO), 2023). Other major types of dementia are vascular dementia, Lewy body dementia, and a spectrum of diseases that contribute to frontotemporal dementia (World Health Organisation (WHO), 2023).

More than 55 million people worldwide have dementia, and about 10 million new cases are diagnosed each year. Dementia is now the seventh leading cause of death and one of the main causes of impairment and dependency in older people (World Health Organisation (WHO), 2023). Dementia affects people in many ways, including physically, mentally, socially and economically. It affects not only people afflicted with the disease, but also their caregivers, families and society (Häikiö et al., 2019). One of the challenges is that a lack of awareness and understanding of dementia can lead to stigma and barriers to assessment, diagnosis, treatment and care (Sideman et al., 2022; World Health Organisation (WHO), 2023).

Atri (2019) observes that the UK is facing an epidemic of dementia. Recent data suggest that there are 944,000 people in the UK with dementia (Dementia Statistics UK, 2022), this figure is expected to increase to 2 million by 2050 (Wittenberg et al., 2019). Furthermore, an estimated 42,000 people living with dementia have early-onset dementia, where the condition is diagnosed before the age of 60 (Wright and McKeown, 2018). In the UK, it is estimated that 60% of patients receiving home care services have a diagnosis of dementia; 25% of hospital beds are occupied by dementia patients, moreover, about 69% of nursing home residents have dementia (Dementia Statistics UK, 2022). Etkind et al. (2017) further highlight that the number of dementia patients requiring end-of-life care is expected to quadruple by 2040. The cost of dementia care to society in the UK is estimated to be £34.77 billion per year, and this cost is projected to rise sharply as the population ages (Public Health England, 2019). However, chronic underfunding in dementia care renders the care system inadequate to meet the needs of patients which is unfair to dementia patients and their families (Alzheimer's Wittenberg et al. 2019). The foregoing demonstrates a

significant problem posed by the ever-increasing dementia care needs, as well as a clear indication of an increasing nursing care burden requiring specialised skills. To mitigate against the dementia epidemic, it is prudent for policy makers to consider creating a stand-alone pre-registration branch of dementia nurse education to future-proof dementia nursing care.

## Addressing the deficits in dementia education and training

It is estimated that 77% of people with dementia also have other comorbidities, the most common being transient ischaemic attack, Parkinson's disease, coronary heart disease, hypertension, diabetes, anxiety and depression (Public Health England, 2019). Also, behaviours that challenge are a common characteristic of dementia. It is estimated that 92% of patients with dementia present with this trait (Kwon and Lee, 2021). All of this suggest that the care needs and management of individuals with dementia are multi-layered and extensive (Grand et al., 2011). Yet, there is a large and growing body of evidence suggesting that many nurses lack the necessary skills and knowledge in dementia care (Royal College of Psychiatrists (RCPsych), 2011; Surr et al., 2017).

The U.K. Higher Education for Dementia Network (HEDN) has been a leading force in the early development of dementia education for pre-registration health and social care professionals over the past two decades, and it continues to play a significant role in the development of dementia education research and pedagogical development within Higher Education Institutions (HEIs) (HEDN, 2022). However, historically, the evolution of organic and localised approaches to dementia care (Hibberd, 2011), poor pedagogical approaches and a lack of standardised education and training in dementia care (Smith et al., 2019) may have contributed to dementia patients being underserved in terms of their mental health, physical health and social care (Cooper et al., 2017). This finding adds weight to our call for the establishment of pre-registration nursing programmes in dementia care.

## Reports and commissions into nursing education in the UK

It is surprising that, despite numerous reports and commissions into nursing education, such as the Briggs Report in 1972 (Chapman, 1973) and the Willis Commission (2012), no recommendations were made for nursing to offer pre-registration education in dementia care. The Willis Commission (2012), which was tasked with gathering evidence on the best methods of delivering consistent high-quality nursing education in the UK, did not foresee nor future proof the dementia pandemic by proposing a specialist field of dementia nursing care. This is particularly concerning given the growing demands of an ageing population and the increasing need for specialised nursing care for dementia patients (Crowther et al., 2013; Etkind et al., 2017).

The Briggs Report was a landmark document that led to the formation of the Nurses, Midwives and Health Visitors Act 1979 (Chapman, 1973). Tierney (2022) points out that the Briggs Report was a 'breakthrough' for nursing education and research in the UK, as it concluded that nursing training in the UK was insufficient to meet the needs of modern society. This led to sweeping changes in professional nursing education. We are now at another crossroads in the history of nursing education. The ageing population is putting increasing pressure on the health and social care system, and the demand for dementia care is growing. To meet this demand, we need to ensure that all nurses have the skills and knowledge to care for people with dementia. One way to do this would be to offer pre-registration education in dementia care. This would ensure that all nurses are equipped to provide high-quality care to people with dementia, regardless of their setting. It would also help to raise awareness of dementia and its impact on individuals and their families.

Policy makers for nurse education have always been open to changes in response to the health needs of the population (Health Education England (HEE), 2015). For example, following the Briggs Report in 1972, professional nursing gained traction and the four branches of nursing were established in 1983: mental health, children, learning disability, and adult. Diploma level nurse training was transferred from hospital-based schools to colleges and universities beginning in 1986, and all nursing courses in the UK became degree level in 2009 (Thomas, 2016). Blane (1991) observes that nursing evolved into four distinct characteristics: It became a highly skilled sector with a well-defined body of specialised knowledge; it established a monopoly over practitioner regulation; it became autonomous to organise, develop and define professional responsibility; and it had a code of ethics that prohibits patient exploitation and professional accountability. Yet, when the Nursing and Midwifery Council (NMC) recently published its pre-registration nursing educational standards (NMC, 2018), it did not recognise the usefulness of a standalone branch for dementia nursing. Remarkably, the NMC (2018) claims that nurses should be competent in providing care for patients with complex dementia needs, but they do not elaborate on how these nurses are to be trained. This is despite the evidence that pre-qualification dementia education can improve knowledge, patient care and outcomes (Rahman and Dening, 2016; Williams and Daley, 2021). Some mental health nursing academics have even gone as far as to say that the NMC's standards are too generic and an attack on nursing specialism (Connell et al., 2022; Mental Health Nurses Academic UK, 2023). We feel that this is a missed opportunity, as dementia is a growing public health challenge and there is a need for more specialist dementia nurses. We concede, however, that due to the variation and heterogeneity in the design of nursing courses in various academic education institutes (AEIs), any specialised dementia nursing education is likely to confront a global challenge of standardisation.

## Learning from learning disability nursing

In the UK, more than 1.5 million people have learning disabilities. It is estimated that learning disability affects 2.16% of adults and 2.5% of children (Mencap, 2022). Today, while many people with learning disabilities live independently, some with specialist support, learning disability generally, has unpleasant roots. Many individuals with learning disabilities were segregated from their communities around the turn of the 20th century. They lived in outdated, inappropriate workhouses or large institution referred to as 'asylums', far from their families and society (Gates, 2022). However, over time, attitudes and policies affecting the lives of people with learning disabilities and those who work with them have shifted. Towards to end of the 20th century, things began to improve with move from institutional care to residential care and 'ordinary living'. The 21st century began with increased governmental concern for the inclusion and empowerment of people with learning disabilities. Policies have emphasised the importance of protecting people with learning disabilities from crime, abuse and exploitation

as their freedoms and community presence grow (Williams, 2013; Gilbert, 2014).

Despite its unpleasant roots, one notable aspect is that nursing care for people with learning disabilities was available much earlier. Stephenson (2019) discusses that the first 'mental deficiency nursing' accreditation was given out by the Medical Psychological Association in 1919, and training for learning disability nurses was subsequently brought in line with training for other branches after the formation of NHS in 1948. Many patients abuse scandals and the eugenics movement, according to Gates (2022), predominated learning disability practice for a long time. However, learning disability nursing practice has steadily changed away from control and towards compassionate and empowering care.

In summary, learning disability nursing has a long history and has developed significantly over time. There are similarities between learning disability nursing and dementia care, such as the need for specialist knowledge and skills. By drawing on the experience of learning disability nursing, we can help to ensure that the future of dementia care is strong and sustainable.

## Pre-registration nursing education: The route to quality dementia care

Our rallying cry is that all pathways to quality dementia care must be through pre-registration nursing education. Although neither recontextualising nor repudiating care of dementia patients is our intention. However, we are concerned by studies that show that the emotional and physical health needs of dementia patients are not being met (Surr et al., 2017; Ganga, 2022). A plausible explanation is that most of the dementia care workforce is unqualified, underpaid, and has no clear career path (Surr et al., 2017). The dire state of dementia care is demonstrated in a systematic review by Aledeh and Adam (2020), who found that dementia carers fall into two categories: often isolated female informal/family carers providing care at home, and inadequately trained formal/professional carers. It is sufficient to argue, albeit cautiously, that there are pockets of prehistoric era to 18th century care described above for some of today's dementia patients. While we recognise that family and other unpaid carers make significant contributions to the support of people with dementia, investing in pre-registration nursing education to train dementia experts will ensure that people with dementia and their families receive high-quality care (Wittenberg et al., 2019). As discussed above, our primary goal is for universities to offer accessible specialist dementia training to help address the shortage in dementia care. Our proposal is opposite to the current trend in which a significant amount of dementia-training is provided in-house, by private providers, or in further and higher education institutions who offer post-graduate training (Surr et al., 2017). Our argument is that these courses do not necessarily equip graduates with clinical or practical skills to manage and coordinate care of dementia patients.

## Contribution to global research and learning

Many countries are facing a shortage of specialist dementia nurses (World Health Organisation (WHO), 2022). Therefore, this article provides an important thought for countries to consider a stand-alone pre-registration branch of dementia nurse education. This would lead to a more specialised workforce with the skills and knowledge to provide high-quality care to people with dementia.

## Conclusion

In conclusion, this article has argued that nursing training that specialises in dementia care should be at the epicentre in pre-registration nursing education. As is the case with most embryonic ideas, it is common for opposing or complementary views to emerge quickly. We acknowledge that any opposing view is a valuable critique to advance nursing education. We can identify at least two opposing views to our proposal. One opposing view might be that we are trying to recontextualise or repudiate already existing good training programmes for dementia. Critics may point to the expert group-led national framework for dementia programmes commissioned and funded by the Department of Health and developed in collaboration by Skills for Health and HEE (Skills for Health, Health Education England, and Skills for Care, 2018). However, critics point out that such programmes may have been volume-driven rather than quality or efficacy (Surr et al., 2017). Furthermore, the framework has been criticised for failing to consider the pedagogical impact of the training, as well as the literacy, numeracy and English language competency of the diverse workforce (Smith et al., 2019). Another opposing view might be that ideas contained in this article are not new, and that we are seeking to duplicate the good work being done by Dementia UK and Admiral Nurses who provide ongoing professional and practical development, and specialist dementia care (Lyons, 2020; Dementia UK, 2023). However, Admiral Nursing is a service model and not an educational model. In fact, Admiral Nurses are offered additional post registration training to support nurses who are interested in dementia care in their practice. Our contention is that dementia nursing education and care must move beyond the exclusive and limited-service model.

This article advocates for policymakers, universities, HEE and NMC to collaborate in the development of pre-registration programmes in dementia care to produce a specialist workforce. We believe that this is essential to ensure the provision of high-quality specialist nursing care to dementia patients. We are optimistic that the partnership between HEE and NMC will be pivotal in advancing this proposal. Further research is needed to explore the perspectives of nursing academics, nurses, practitioners and carers on this proposal.

**Open peer review.** To view the open peer review materials for this article, please visit http://doi.org/10.1017/gmh.2023.80.

**Data availability statement.** Data availability is not applicable to this article as no new data were created or analysed in this study.

**Acknowledgements.** We are grateful to the constructive feedback from the anonymous reviewers.

**Author contribution.** I.T. and G.G. made equal contributions to the conception, drafting and revising of the manuscript. Both authors have read and agreed to the final version of the manuscript and agree to be accountable for all aspects of the work.

**Financial support.** This article has no grant from any funding agency, commercial or not-for-profit sectors.

**Competing interest.** The authors declare no competing interests.

**Ethics statement.** This article did not require ethics approval as no data were collected from participants.

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
