## [Reviewer Report]

Thank you for submitting this article which poses an interesting argument about the role of higher education in improving the delivery of dementia care specifically through a proposed branch of nursing in dementia. I can see there is merit in the development of a consistent approach to education in dementia through developing a distinct branch which may allow for national regulatory standards such as those prescribed for learning disability or other branches. However there are a number of assumptions made which I would question and a lack of acknowledgement of the contribution of other professional groups and systems which significantly influence the context and ability to improve delivery of dementia care.

For example, it is argued that the lack of specialist dementia nurses is one of the reasons why policy initiatives to improve dementia care may be ‘derailed’. Although this may be a laudable argument, nurses work within a complex system and many other roles and factors will influence the delivery of care. Whilst nurses make a significant contribution to the delivery of care for people with dementia, developing a specific branch for nursing in dementia does not address the challenges of the workforce, paucity of specialist roles and workplace settings. Are you suggesting that those who have completed the training would work in dementia specific environments and if so where would these be?

There are repeated references to the need to improve dementia education at pre-registration level and references made to identify the lack of skills and knowledge. However some of these references are misrepresented e.g. RC Psych (2011) refers to an audit of healthcare staff in general acute hospitals and not healthcare settings in general. It would also be helpful to reference some of the work about the delivery of dementia education in pre-registration training e.g. Pulsford et al 2007;https://www.sciencedirect.com/science/article/abs/pii/S026069170600027X;

Tullo & Gordan 2013, https://bmcgeriatr.biomedcentral.com/articles/10.1186/1471-2318-13-29;

Collier et al 2015, https://www.sciencedirect.com/science/article/abs/pii/S0260691715000866;

Williams & Daley 2021, https://www.sciencedirect.com/science/article/abs/pii/S0260691720315926

A claim is made that the inclusion of a standalone branch for dementia in nursing would maintain quality assurance. Whilst I admire the aspiration this is a huge claim to make and I would suggest very difficult to prove. The heterogeneity of professional education across different is already a challenge across different institutions and this should be acknowledged. There is also no mention of the recent NMC educational standards which are significantly shaping the way in which pre registration nursing education is being delivered. How do you think a new branch would align with this?

The critique about Admiral Nursing seems to conflate the arguments about nurse education and service delivery and as such is somewhat confusing. Admiral Nurses as dementia nurse specialists are offered additional post registration training to support them in their practice. This is not an ‘educational model’ but a highly critical comparison is made directly between Admiral Nursing and suggestion that it works against the delivery of dementia education in pre-registration nursing. This argument does not follow.

There are some useful points made about dementia not fitting in both mental health and ‘older’ adult nursing specialties. However older adult nursing is surely a service delivery model as opposed to an educational model. Dementia is a degenerative neurological condition and this should be referenced appropriately e.g. Dementia Friends 2017 reference here is insufficient.

Lastly I would encourage the authors to explore the work that has previously been carried out about improving the delivery of pre-registration education e.g. UK Higher Education for Dementia Network https://hedn50472928.wordpress.com/

---

## [Reviewer Report]

I believe that this article presents a thought provoking argument for establishing a firmer foundation of pre-registration dementia education. I agree that as educators it is in our remit to improve the education we offer to pre-registration students in order to eventually improve patient care outcomes. The arguments you make here for a RN-Dementia registration is interesting.

Whilst I am always happy to see a section on the history of nursing I was not fully clear on the arguments links to dementia nursing. I understand that it was leading to the learning from Learning Disability Nurses, but I felt that this area could be developed a little more.

As a minor amendment I believe Admiral Nurse should be capitalised throughout as this is a nursing role- it is referred to in this way on the Dementia UK website and what my colleagues who are Admiral Nurses would sign as.

---

## [Reviewer Report]

This paper is an opinion piece on proposing a new training curriculum for dementia care nurses. It is largely focused on the UK. There are no references on how this relates to other global research and other global learnings in this area. Please include a section on this.

---

## [Reviewer Report]

Dear Editor,

We are writing to follow up on the recent submission of our manuscript entitled “`”. We have carefully considered the feedback from the two reviewers and have made the following changes to the manuscript:

• We have included the abstract, social media summary, graphical abstract, and impact statement as requested.

• We have corrected the references that were misrepresented.

• We have added references to the work on the delivery of dementia education in pre-registration training.

• We have clarified the argument about Admiral Nursing and its relationship to the delivery of dementia education in pre-registration nursing.

• We have referenced dementia as a degenerative neurological condition.

• We have added a reference to the work of the UK Higher Education for Dementia Network.

• We have capitalized “Admiral Nurse” throughout the manuscript.

• We have deleted the section on history of nursing to make our argument more focused,

We believe that these changes have addressed the concerns raised by the reviewers and have strengthened the manuscript. We would be grateful if you would consider the revised manuscript for publication.

Sincerely,

Dr Isaac Tuffour

Dr Griffin Ganga

---

## [Reviewer Report]

I enjoyed this article and the points it was making. It has given me some thought as I progress the development of our own pre-registration curriculum.

thank you for the opportunity it has given.

---

## [Reviewer Report]

1. Please provide a detailed abstract of the paper. Include more information on what is discussed in the rest of the paper.

2. Please address the reviewer’s comment: Dementia is a degenerative neurological condition and this should be referenced appropriately. The two references that have been added are insufficient. 

3. In your cover letter and manuscript please indicate how you have responded to these comments from the reviewer: 

i. The critique about Admiral Nursing seems to conflate the arguments about nurse education and service delivery and as such is somewhat confusing. Admiral Nurses as dementia nurse specialists are offered additional post registration training to support them in their practice. This is not an ‘educational model’ but a highly critical comparison is made directly between Admiral Nursing and suggestion that it works against the delivery of dementia education in pre-registration nursing. This argument does not follow. 

ii. There are some useful points made about dementia not fitting in both mental health and ‘older’ adult nursing specialties. However older adult nursing is surely a service delivery model as opposed to an educational model

---

## [Reviewer Report]

Dear Editor,

We are writing to follow up on the recent submission of our manuscript entitled “`”. We have carefully considered the feedback from the two reviewers and have made the following changes to the manuscript:

• All changes are in green font.

• We have reviewed the abstract, social media summary and impact statement as requested.

• A graphical abstract figure is provided separately.

• We have corrected the references that were misrepresented.

• We have added references to the work on the delivery of dementia education in pre-registration training.

• We have clarified the argument about Admiral Nursing and its relationship to the delivery of dementia education in pre-registration nursing.

• We have included additional references of dementia as a degenerative neurological condition.

• We have added a reference to the work of the UK Higher Education for Dementia Network.

• We have capitalized “Admiral Nurse” throughout the manuscript.

• We have reviewed the critique about Admiral Nursing. We acknowledge that Admiral Nurses are already qualified nurses who undergo specialist post-qualification training. We do not argue that Admiral Nursing is an educational model.

• We have strengthened the difference between older adult nursing as a service delivery, and pre-registration RN-Dementia programme as educational model.

• We have deleted the section on history of nursing to make our argument more focused,

We believe that these changes have addressed the concerns raised by the reviewers and have strengthened the manuscript. We would be grateful if you would consider the revised manuscript for publication.

Sincerely,

Dr Isaac Tuffour

Dr Griffin Ganga

---

## [Reviewer Report]

Please respond to the following comments that were sent in the previous comments.

In your cover letter please indicate how you have responded to these comments from the reviewer: 

i. The critique about Admiral Nursing seems to conflate the arguments about nurse education and service delivery and as such is somewhat confusing. Admiral Nurses as dementia nurse specialists are offered additional post registration training to support them in their practice. This is not an ‘educational model’ but a highly critical comparison is made directly between Admiral Nursing and suggestion that it works against the delivery of dementia education in pre-registration nursing. This argument does not follow. 

ii. There are some useful points made about dementia not fitting in both mental health and ‘older’ adult nursing specialties. However older adult nursing is surely a service delivery model as opposed to an educational model

---

## [Reviewer Report]

There are no responses to these comments from the reviewer in the cover letter provided:

In your cover letter please indicate how you have responded to these comments from the reviewer: 

i. The critique about Admiral Nursing seems to conflate the arguments about nurse education and service delivery and as such is somewhat confusing. Admiral Nurses as dementia nurse specialists are offered additional post registration training to support them in their practice. This is not an ‘educational model’ but a highly critical comparison is made directly between Admiral Nursing and suggestion that it works against the delivery of dementia education in pre-registration nursing. This argument does not follow. 

ii. There are some useful points made about dementia not fitting in both mental health and ‘older’ adult nursing specialties. However older adult nursing is surely a service delivery model as opposed to an educational model